# Effect of Walking on Sand with Dietary Intervention in OverweightType 2 DiabetesMellitusPatients: A Randomized Controlled Trial

**DOI:** 10.3390/healthcare8040370

**Published:** 2020-09-29

**Authors:** Mohamed Seyam, Faizan Kashoo, Mazen Alqahtani, Msaad Alzhrani, Fahad Aldhafiri, Mehrunnisha Ahmad

**Affiliations:** 1Department of Physical Therapy & Health Rehabilitation, College of Applied Medical Sciences, Majmaah University, Al Majmaah 11952, Saudi Arabia; m.seyam@mu.edu.sa (M.S.); mm.alqahtani@mu.edu.sa (M.A.); m.alzhrani@mu.edu.sa (M.A.); 2Public Health Department, College of Applied Medical Sciences, Majmaah University, Al Majmaah 11952, Saudi Arabia; f.aldhafiri@mu.edu.sa; 3Department of Nursing, College of Applied Medical Sciences, Majmaah University, Al Majmaah 11952, Saudi Arabia; m.ahmer@mu.edu.sa

**Keywords:** diabetes mellitus, walking, rehabilitation

## Abstract

(1) Background: The primary goal of this study was to assess the effect of sand walking on Hemoglobin A1c (HbA1c), Body Mass Index (BMI), waist circumference, and quality of life among individuals with Type-2 Diabetes Mellitus (T2DM). (2) Methods: A randomized-controlled design was conducted on 66 overweight participants suffering from T2DM. Participants were randomly allocated to sand walking (SW) (*n* = 33) and normal walking (NW) (*n* = 33) groups. Participants performed moderate-intensity walking for 30 min, 3 times a week for 4 months. Participants walking on sand had statistically significant mean scores for HbA1c, BMI, waist circumference and quality of life((M = 7.32, SD = 0.47),(M = 25.77, SD = 1.366),(M = 92.94, SD = 2.59), (M = 91.48, SD = 34.08)) than those walking on leveled surface ((M = 8.38, SD = 0.77),t(52.8) = −6.73, *p* = 0.003, (M = 26.80, SD = 1.38), t(64) = −3.05, *p* = 0.001,(M = 98.12, SD = 2.16.3), t(64) = −3.75, *p* = 0.001, (M = 112, SD = 33.7), t(64) = −2.45, *p* = 0.017)respectively. (3) Conclusions: Regular SW with a healthy dietary regime for 4 months led to a statistically significant difference in HbA1c, BMI, waist circumference, and quality of life as compared to NW group.

## 1. Introduction

Type 2 diabetesmellitus (T2DM) is one of the prevalent metabolic disorders worldwide [1]. Its prevalence has increased worldwide in the last three decades due to immobility, unhealthy dietary habits, environmental factors, and individual genetic vulnerability [2]. Diabetes mellitus is a global epidemic estimated to have affected 415 million people in 2015 and is estimated to reach 642 million by 2040 [3]. Developing countries like the Kingdom of Saudi Arabia (KSA), Bahrain, Kuwait, Qatar, Lebanon, and the United Arab Emirates are among the world’s top countries with the highest prevalence of T2DM affecting 32.8 million people in 2011, expected to reach 60 million by 2030. In KSA, dietary habits [4], genetic predisposition, and sedentary lifestyle are the main cause of early-onset T2DMx [5].

Uncontrolled T2DM reported to cause multiple complications and alter the normal physiological process of the body. Prompt dietary management [6] and regular exercise are reported to improve blood glucose homeostasis and eventually improve the quality of life [7]. Physical activity such as regular walking is associated with multiple physical and psychological benefits [8]. Walking and a healthy diet is a cornerstone in the treatment of T2DM. Several studies have shown positive short-term effects of walking in T2DM [9,10]. Combination of walking and weight training is reported to cause cardiovascular adaptations, muscle hypertrophy, increased capillary density in the muscles of patients with T2DM as well as in healthy people [11]. Furthermore, 30 min of brisk walking is reported to improve blood glycemic metabolism and reduces cardiovascular risk factors, such as high blood pressure, lipid disorders, and fat mass buildup [12]. However, adults with, mild to moderate arthritic changes especially in knee joints make walking a bit more challenging. Sand walking would be a better alternative than firm surface walking, which is reported to produce less joint reaction forces. To the best of our knowledge, scientific literature lacks evidence about the effect of sand walking combined with individualized dietary intervention inT2DM. Therefore, the purpose of this study was to compare the effect of sand walking (SW) as compared to normal walking (NW) on glycemic metabolism, weight, and quality of life in T2DM. We hypothesize that there will be significant difference in better health benefits between participants in SW and NW group.

## 2. Materials and Methods

### 2.1. Design

We conducted a 16-week, multi-center, randomized controlled trial. Eligible participants were randomly assigned to sand-walking and normal walking group. The participants and therapists could not be blinded due to the apparent difference in the intervention in the two groups, but the therapists responsible for assessing outcome variables were blinded to the allocation of participants. The study protocol was reviewed and approved by the Deanship of Scientific Research at Majmmah University (MUREC- Oct. I 6/COM-201 8/5). Participants gave written informed consent before being enrollment in the study. The study began in January 2017 and was completed in September 2019.

### 2.2. Setting

We recruited participants by searching the records databases of the hospital and community health centers and by direct advertising. We also recruited participants from the outpatient physical therapy department of University affiliated Hospital (Figure 1). The Clinical trial was registered with the US clinical Trial Registry No. NCT04364685.

### 2.3. Eligibility

The inclusion criteria wereT2DM for more than one year, HbA1c ranging from 7% to 10%, BMI between 25–30 kg/m^2^, age in the range of 40 to 75 years, and stable anti-diabetic treatment for the last 6 months and sedentary level of activity (activities restricted to activities of daily living). Participants with a history of recent myocardial infarction, diabetic neuropathy, using walking aids, and unstable angina, severe musculoskeletal disorders that make walking difficult, current cigarette or other forms of smoking, and severe respiratory disease were excluded. A comprehensive physical exam and counseling session were performed before the start of the group exercise session. Each participant was explained about the diet regime Appendix A recommended by a dietician. The participants reported their rate of perceived exertion on the 20-points Borg scale at the end of each 30-min walking session.

Acclimatization to exercises: Every eligible participant participated in a preparatory exercise 2 to 3 weeks before the commencement of the study. The exercises included walking on a firm surface for 10–20 min 3 times a week. Participants who were not regular (*n* = 1) or showed an abnormal cardiovascular response to exercise (*n* = 1) were excluded.

### 2.4. Randomization

Computer-generated random numbers were assigned to each of the eligible participants. These random numbers were printed on a paper separately and folded in a jar. A second person, who was not part of the study and neither was aware of the study protocol, picked these folded papers from the jar one by one and sorted them into two groups. The third person who was also not part of the study named the two groups of papers as control and experimental.

Procedure: The study consists of two groups NW (*n* = 33), SW (*n* = 33). Each intervention lasted for 45 min, which consisted of 10 min of warm-up, 30 min walking intervention, and 5 min cool down. All the groups had to perform moderate-intensity walking, 3 times a week for 16weeks (4 months). Three physiotherapists supervised the treatment session. All sessions were scheduled in the morning. The physiotherapists involved were randomly assigned to supervise any of the two groups. To reduce attention bias all the participants were attended with equal enthusiasm. To effectively manage the risk of the hypoglycemic episode during or after the walking session, appropriate precautions were taken.

Participants in the NW group (*n* = 33) performed outdoor walking. A group walking session of moderate-intensity training for 30 min was conducted at the University Track and Field ground wearing their regular comfortable sports shoe. Vo2 max was not calculated during the training session, but the moderate intensity of exercise was reported on Borg rating for perceived exertion scale (level 13–14) [13]. Each participant was evaluated and interviewed by the fourth blinded physiotherapist not involved in the intervention, at the recruitment, and at the end of the 16th week.

Participants in the SW group performed walking on soft sand. Twenty-meter-long leveled soft sand was available within the vicinity of the university campus. The pathway of sand was primarily for annual university athletic meet. The sand was examined by three therapists for any sharp stones and was leveled after every session. The consistency of sand was such that the average weighted individual feet would easily be covered with sand up to the ankle joint. The patients wore their normal shoes with comfortable clothing. After every 10 min of walking, the therapist communicated with the participants to enquire about their state of exertion. Participants could take rests but were required to complete 30 min of walking using a stopwatch. All the participants took two to three rest periods before completing a full 30 min walking session.

Outcome measures: HbA1c, BMI, waist circumference, and quality of life were calculated at baseline and on the 16th week.

A laboratory assistant obtained a blood sample from the participants. HbA1c was analyzed using high-performance liquid chromatography. The normal range of HbA1c ranges from 4.5% to 6.3%. The lipid profile was also measured by a standard method.

Bodyweight and height were measured by a standard weighting machine (Electronic BMI measuring height-weight scale machine-SGC200120, (Yiwe Sangong Fitness equipment Factory, Zhejiang, China). A standard inch tape was used to measure the waist circumference at the level of the posterior superior iliac spine.

Achieving dietary change requires strong will power and giving up long traditional patterns of eating practice. A dietician interviewed each participant to formulate a diet chart tailored to their body type and level of activity. The intention was not to reduce the weight but to make sure that the dietary intake was similar across the groups. Participants in the experimental and control groups showed a significant amount of compliance with the training and diet regime. Besides, the participants were instructed to remain active in their daily activities after the session.

Health-related quality of life was evaluated by the Arabic version of Diabetes 39 (D-39). D-39 includes thirty-nine questions with a total score of 253. Each question is rated on a Likert scale ranging with a score of 1 to 7. Therefore, the minimum score is 39, indicating the worst quality of life and the highest score is 253, indicating the highest level of quality of life. The Arabic version of the D-39 has been validated.

Dietary advice includes carbohydrates in the form of starch, avoiding refined sugar and usage of non-nutritive sweeteners, replacing food containing saturated fatty acids, salts consumption reduced to a minimum, diet with high quality of protein, small multiple meals over a day to avoid post-prandial peak blood sugar level. Other important instructions related to the type of food are provided in detail in the Appendix A.

## 3. Results

### 3.1. Statistical Analysis

The power analysis was performed by retrieving the data from a similar study [14]. To get statistical power of at least 95% (2-tailed α= 0.05), the minimum number of participants in each group must be 30. The total number of participants in this study was 69. Estimating a possible drop rate of 10%, 66 participants were recruited for this study.

We conducted all analyses using the intent-to-treat principle. We conducted a descriptive analysis to obtain the missing value, revealed 3% (*n* = 2), 1.5% (*n* = 1), and 1.5% (*n* = 1) of missing value from pre-test evaluation of BMI, pre-test evaluation of HbA1c, and post-test of BMI, respectively. Little’s Missing Completely at Random (MCAR)test was performed on the data set to analyse the missing values that are randomly missing (Little’s MCAR test Chi square = 28.663, DF = 24, Sig = 0.233). The results reported that the values missing are missing at random. Expectation maximization was used to replace the missing values separately for each variable, later the files were merged as one for analysis.

Our randomized controlled trial consists of 66 participants randomly allocated into experimental (sand walking, *N* = 33) and control group (Normal walking, *N* = 33). By comparison, the sand walking group is associated with better improvement in BMI, HbA1c, waist circumference, and quality of Life. To test the hypothesis that the sand walking was associated with statistically significantly different mean dependent variables, a student *t*-test was performed to test between-group (pre-test control with pre-test experimental) and paired sample *t*-test to analyses within-group mean difference (pre-test control with post-test control). To meet the assumption of the test, normality and homogeneity test was performed. All the variables met the assumption of normality; however, Post-test results of HbA1c failed the homogeneity of variance via Levine’s F test, F (11.170), *p* = 0.001. Therefore, the test output for the Post-HbA1c variable was interpreted as sphericity not assumed. (F = 45.176, df = 52.872, *p* = 0.001). All the other variables satisfied the assumptions and the post-test score was statistically significant in the experimental group. A graphical representation of means and the standard error of the mean is displayed in Figure 2 and Figure 3.

We randomly selected 66 participants suffering from chronic T2DM, sedentary, and overweight. The participants were allocated randomly into two groups, Firm-surface walking (*n* = 33), Sand Walking (*n* = 33).

### 3.2. Chi-Square Test (Categorical Data Analysis)

A chi-square test of independence showed that there was no significant difference between gender distribution in the total sample, X^2^ (1, *N* = 66) = 1.515, *p* = 0.218. However, when analyzed with independent sample, *t*-test male participants(M = 93.50, SD = 2.72) had significant reduction in waist circumference compared to female (M = 94.79, SD = 2.31), *t*(64) = −2.015, *p* = 0.048.The baseline analysis of variables in the NW and SW groups such as age, gender, HbA1c, BMI, waist circumference, and quality of life were not statistically significant (Table 1).

### 3.3. Within-Group and between-Group Analysis

The independent sample *t*-test demonstrated a statistically significant mean difference between the experimental group (Sand walking) as compared to control group (Normal walking) for HbA1c (*t*(52.8) = −6.73, *p* = 0.003), BMI (*t*(64) = −3.05, *p* = 0.001), waist circumference (*t*(64) = −3.75, *p* = 0.001), and quality of life (*t*(64) = −2.45, *p* = 0.017) (Table 1). The within-group pre-test and post-test analyses of each variable were performed by paired sample *t*-test for HbA1c, BMI, waist circumference, and quality of life. There is a statistically significant difference in the mean scores of dependent variables before and after the intervention in both the groups (Table 2).

## 4. Discussion

Our study aimed to evaluate the effect of sand walking on HbA1c and other health indicators in people with T2DM. There was significant mean change in HbA1c (1.3%), BMI (0.9 kg/m^2^), waist circumference (5.3), and Quality of life (66.2) in SW group as compared to HbA1c (0.5%), BMI (0.8 kg/m^2^), waist circumference (3cm), and Quality of life (50.3) in NW group.

The change in HbA1c in NW group (0.5%) was similar to a recent study reported that the combination of recommended diet and exercises will result in a better reduction (0.041%) in the elevated HbA1c [15]. However, the significant reduction of 1.3% in HbA1c in SW group could be due to effect of walking on sand and the typically characteristics of our sample, that is, sedentary, overweight, and chronic T2DM. We used soft sand as a pathway for our experimental group participants to perform walking. Sand could provide an ideal surface, if the patient suffers from increased pain due to increased joint reaction forces or arthritis [16]. Researchers have reported reduced joint reaction forces and increased energy expenditure during sand walking [17]. There is little research available that uses sand as an interface to treat various neuromuscular disabilities or general systemic disorder. One of the studies on stroke [18] and multiple sclerosis [19] uses sand as a pathway to train walking. Both researches have reported improvement in gait parameters. In our study, participants performed 90 min of walking per week, (30 min of walking 3 times a week). A recent review and meta-analysis [20] reported the use of structured exercises must be more than 150 min per week to be associated with health benefits. A duration of exercises of more than 150 min reduced HbA1c by 0.89% as compared to 0.36% when the duration was less than 150 min. Some studies employed a higher intensity of training [21] and included participants with higher BMI and found a significant reduction in HbA1c [22]. It was reported that lifestyle interventions are more efficient among individuals with a BMI higher than 25 kg/m^2^. Higher intensity of exercises among participants with T2DM requires a high level of dedication and effort from the individual.

In the control group, there was a mean change of 0.8 kg/m^2^ in BMI as compared to 0.9 kg/m^2^ in the experimental group. Similar findings were also reported in a study that used low-calorie diet with exercise and found a reduction in waist circumference, BMI, and better glycemic homeostasis among individuals suffering from T2DM [23]. In our study, we used a more pragmatic approach towards dietary control, as the level of activity of our participants was sedentary, overweight, and middle-aged [24,25].

Our study reported a reduction in waist circumference among participants in the SW group of around 5.3 cm as compared to 3cm in participants in NW walking. A recent study has reported that there is a positive relationship between the waist circumference and mortality. Waist circumference is also a valid tool to predict cardiovascular diseases in the future.

Quality of life assessed by D-39 showed moderate improvement in both groups. However, scores were better in SW group as compared to NW group. A similar study reported improvement in mental health and SF-36 score with 9-month resistance training as compared to aerobic and combined training [26]. A recent cross-sectional study also demonstrated the cognitive effect of T2DM. The study concluded that there is a significant association between depression, anxiety, and T2DM [27]. A study examines the relationship between the health-related quality of life and physical activity in T2DM. They found a negative correlation between the quality of life and the level of physical activity. A study by Duviviers et al. [28] reported that 60 min of moderate-intensity exercises are sufficient to overcome the negative effects sitting for the rest of the day. It was also reported that short bouts of sitting showed a beneficial effect on insulin sensitivity and the level of blood plasma [29]. Six-minutes’ walk test has been reported to have a significant correlation with HbAc1 (Relative Precision 1.57, χ^2^ < 0.05).The 6-min walk test is reported to be strong to predict the physical capacity of the individual with T2DM [30].

A prospective cohort study involving 4681 healthy adults conducted between 1981–2016 found that higher muscle strength is associated with a lower risk of development of T2DM [31]. A study reported improvement in glucose and lipid metabolism after light bouts of exercises after prolonged sitting in an individual with T2DM [32]. The sedentary busy lifestyle makes it difficult to adhere to the structured exercise protocol. The main goal of our study was to determine realistic training (superior walking method) combined with dietary restriction. Moreover, sand walking was an enjoyable experience for all the participants and required more effort than normal firm surface walking. The scientific literature has reported that sand walking increases the cost of energy [33], reduces walking velocity [34], improves lower limb strength [35], and increases relative muscular activity to maintain balance [36] as compared to the flat surface. These factors might directly or indirectly influence the change in the level of HbA1c, BMI, waist circumference, and quality of life.

There are limitations in our study. The number of participants in the study was relatively small. Future studies must consider involving large samples to obtain robust evidence. The research included overweight diabetic participants without any associated disorders. Therefore, the results of the study cannot be generalized to the diabetic population with associated disorders. The participants reported self-perceived moderate intensity exertion levels through Borg rating for perceived exertion scale. The level perception of moderate-intensity of exercises might have been overestimated or underestimated by the participants. The speed and distance traveled by the participants were not calculated. Future studies may use gadgets to measure the temporal and spatial characteristics of gait.

## 5. Conclusions

There was a significant change in HbA1c, BMI, Waist circumference and quality of life in the patient with T2DM after 4 months of sand walking training and dietary control.

## Figures and Tables

**Figure 1 healthcare-08-00370-f001:**
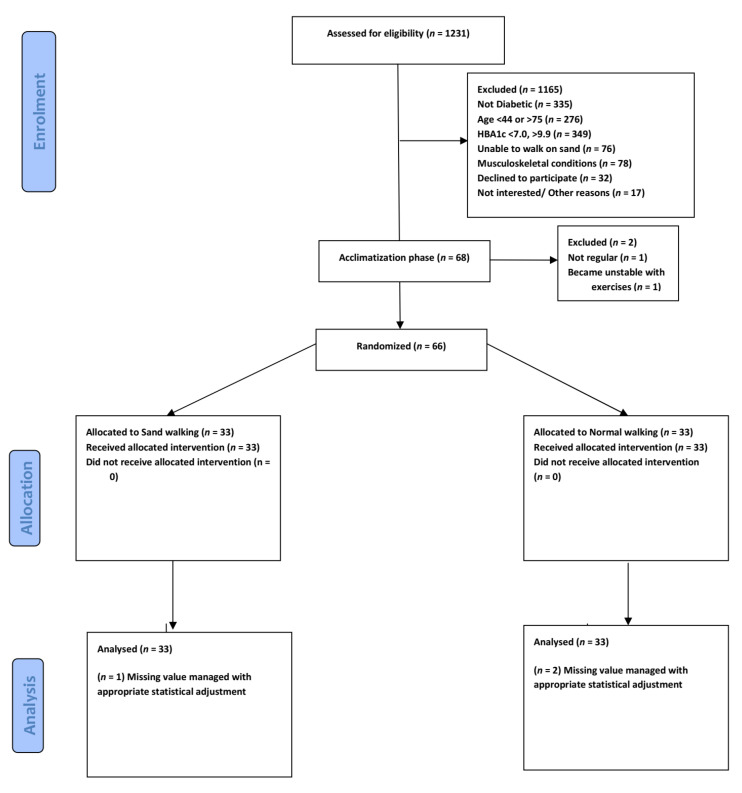
Consolidated Standards of Reporting Trials of participants through enrolment, allocation, and Analysis.

**Figure 2 healthcare-08-00370-f002:**
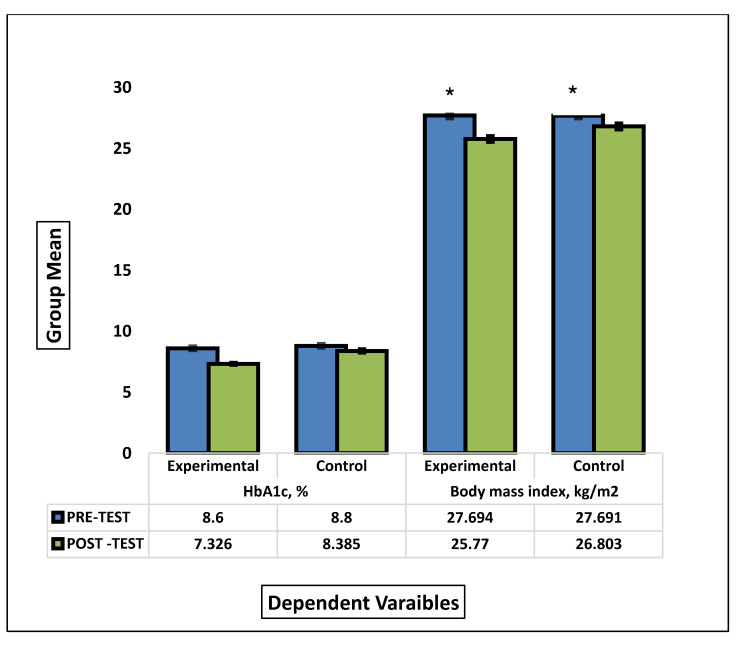
Comparison between the group means of HbA1c and BMI before and after the intervention. * signify statistical significant different between pre-test and post test mean scores of experimental and control groups.

**Figure 3 healthcare-08-00370-f003:**
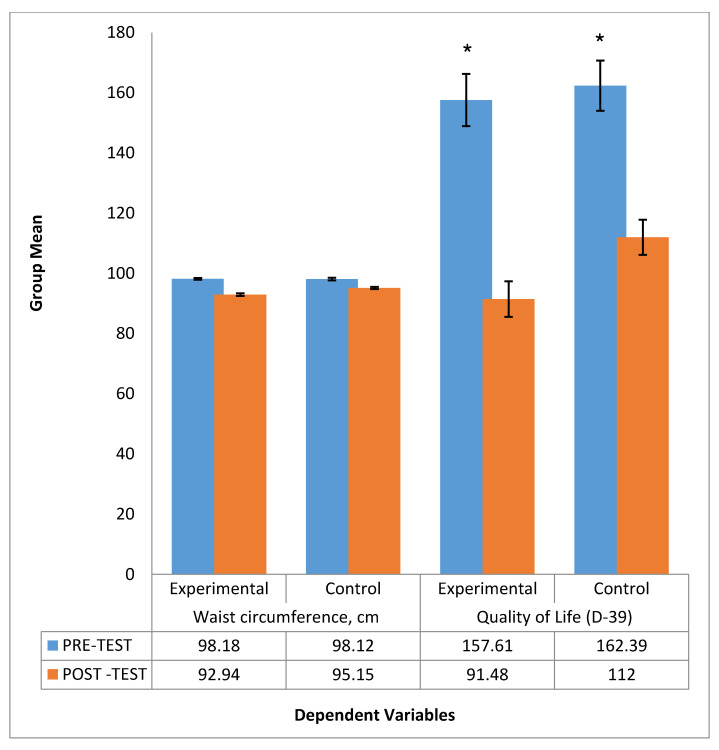
Comparison between the group mean of Waist circumference and Quality of life before andafter intervention. * signify statistical significant different between pre-test and post test mean scores of experimental and control groups.

**Table 1 healthcare-08-00370-t001:** Baseline and post-intervention mean difference between variables.

Characteristics	Group	NW (m ±SD) Baseline	Between Group Difference (*p* *)	SW (m ±SD) Post-Intervention	Between Group Difference (*p* *)
		Pre-Intervention	Post-Intervention		
Males/Females (*n*/*n*)	Experimental	20/13		-	-
	Control	18/15			
Mean age (SD) y	Experimental	52.8 (8.924)	0.713	-	-
	Control	52.1 (7.715)			
HbA1c, %	Experimental	8.6 ± 0.8746	0.355	7.3 ± 0.4706	0.003
	Control	8.8 ± 0.8682		8.3 ± 0.7726	
Body mass index, kg/m^2^	Experimental	27.6 ± 1.2762	0.992	25.7 ± 1.3664	0.001
	Control	27.6 ± 1.2511		26.8 ± 1.3828	
Waist circumference, cm	Experimental	98.2 ± 1.793	0.897	92.9 ± 2.597	0.001
	Control	98.1 ± 1.996		95.1 ± 2.167	
Quality of Life (D-39)	Experimental	157.6 ± 49.801	0.692	91.4 ± 34.089	0.017
	Control	162.3 ± 48.069		112.0 ± 33.706	

NW, Normal walking; SW, Sand walking; m, mean; SD, Standard Deviation; *p* *, independent sample *t*-test.

**Table 2 healthcare-08-00370-t002:** Pre-posttest analysis within the group by Paired sample *t*-test for within-group difference.

Variables	Paired Differences (Control)							
	Mean	Std. Deviation	Std. Error Mean	95% Confidence Interval		*t*	df	*p*
				Lower	Upper			
BMI	0.8	0.8838	0.1538	0.5745	1.2013	5.771	32	0.001
HbA1c	0.5	0.9817	0.1709	0.0670	0.7633	2.429	32	0.021
Waist Circumference (cm)	3	2.215	0.386	2.184	3.755	7.703	32	0.001
QOL	50.3	35.759	6.225	37.714	63.073	8.096	32	0.001
**Variables**	**Paired Differences (Experimental)**							
	**Mean**	**Std. Deviation**	**Std. Error Mean**	**95% Confidence Interval**		***t***	**df**	***p***
				**Lower**	**Upper**			
BMI	0.9	0.5013	0.0873	1.7465	2.1020	22.052	32	0.001
HbA1c	1.3	0.9713	0.1691	0.9292	1.6181	7.532	32	0.001
Waist Circumference (cm)	5.3	3.279	0.571	4.080	6.405	9.184	32	0.001
QOL	66.2	51.849	9.026	47.736	84.506	7.326	32	0.001

BMI, Body Mass Index; HbA1c, Hemoglobin A1c; *t*, *t*-value of the test; df, degree of freedom; *p*, Significance; QOL, Quality of life; m, mean; SD, Standard Deviation; *p*, Paired sample *t*-test.

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
