# Peer review of "Effect of Walking on Sand with Dietary Intervention in OverweightType 2 DiabetesMellitusPatients: A Randomized Controlled Trial"

_healthcare, 2020, doi:10.3390/healthcare8040370_

Round 1

Reviewer 1 Report

Mohamed Seyam et al. investigated the effect of sand walking on HbA1c, BMI, waist circumference and QOL in the patients with T2D. The patients with sand walking demonstrated significant decline in HbA1c, BMI, waist circumference and improvement of QOL compared with those with walking on leveled surface. Although the intervention with san walking is interesting, the mechanism for its effectiveness has not been well-demonstrated.

Major comments

  1. In line 20, what did authors mean in moderate-intensity walking? How did the participants measure the intensity of walking? In NCT04364685, the authors described that the participants would decide the pace of walking.
  2. Inline 60, the outcome was measured after 16-week of intervention. In NCT04364685, the authors described that primary and secondary outcomes will be measured after 2 years.
  3. In lines 73-82, the exclusion criteria should be described. For example, the patients with diabetic neuropathy should be avoided since they may have heat burn, injuries, or infection on sand walking. The authors should include this issue in the limitation of the study in discussion.
  4. In lines 92-114, how did the authors confirm physical activity in both groups were identical. The step meter or physical activity meter should be used.
  5. Although HbA1c, BMI, waist circumference and QOL were improved in the current investigation, the mechanism for the benefits in sand walking is not demonstrated.

Author Response

Dear Reviewer,

Thanks a lot for the valuable comments made on the manuscript. We have replied to each comment point by point in the attached file.

Regards  

Reviewer 2 Report

Title: Effect of walking on sand with dietary intervention in overweight Type 2 Diabetes Mellitus patients: A randomized Controlled Trial.

Summary: The manuscript under review aims to investigate  the effect of sand walking on HbA1c, BMI, waist circumference, and quality of life among individuals with Type-2 Diabetes Mellitus (T2DM).

The idea to compare the health effects of sand walking vs. normal surface walking is novel and intriguing and will most likely be of a high level of interest to the readers. The manuscript is well written, easy to follow and understand with clearly labeled figures with robust data provided and concise, clear content. The study is well designed, methods and statistical analyses are appropriate and up to date. Reference list is comprehensive and complete. Results are clearly explained and support the conclusions of the authors.

A minor suggestion to the authors would be to add a label to the bar graphs (* or

Author Response

(The authors gave the same response as above.)

Reviewer 3 Report

In this manuscript, Seyam et al. conducted a randomized controlled trial investigating effects of walking on sands vs walking on normal path along with dietary intervention on patients suffering with Type 2 Diabetes Mellitus patients. The authors conducted trail on 66 patients and measured changes in HbA1c, BMI, waist circumference and quality of life in these patients. They observed significant changes in these outcomes in patients following healthy diet along with walks in sands.

The manuscript is well written but can not be accepted in its current state. Following changes are warranted:

1) For the figure:2, and 3; the authors should change the color of pre-test vs post-test data for better visualization. Also, they should include p values and * signs to show if the results are significant or not.

2) This randomized controlled trial included 66 patients, 33 patients for each group. It is a very small sample size to draw a conclusive for this kind of study. The authors should address this concern.

3) The authors should provide some background of outcomes such as HbA1c and significance of such factors to choose among all other factors.

Author Response

(The authors gave the same response as above.)

Round 2

Reviewer 1 Report

Major comments

  1. The evaluation of intensity of walking is necessary.
  2. The authors should not change the protocol after the submission of the manuscript.
  3. The exclusion criteria should be fixed before the initiation of the trial.
  4. The authors should confirm the physical activity in 2 groups is identical.
  5. The mechanism for the benefits in sand walking is still not clear.

Author Response

Dear Respected Reviewer,

Thank a lot for the valuable comments on our paper. we appreciate your time and effort. we have tried to answer your comments on our paper point by point and in detail.

Regards 
